# Assessing Public Service Distribution in Abha and Bisha Cities, Saudi Arabia: A Comparative Study

Khaled Ali Abuhasel 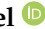

Mechanical Engineering Department-Industrial Engineering Program, College of Engineering, University of Bisha, Bisha 61922, Saudi Arabia; kabuhasel@ub.edu.sa

**Abstract:** The research below aims to examine the spatial distribution and efficiency of public services in Abha and Bisha. Abha is the capital of the Asir region and had a population of 446,697 people in 2022, while Bisha had a population of 248,452 people in the same year. Both cities have their unique geographical features, such as valleys, dams, and agricultural significance. This study utilizes spatial modeling and statistical analysis to analyze data collected via a questionnaire administered to the residents of these cities and formulates several hypotheses to guide the research, including hypotheses related to differences in public services based on gender, age group, and citizenship status. To analyze the data, a combination of analytical descriptive approaches, including statistical methods conducted with SPSS software and geographic information system (GIS) techniques using ArcGIS software were employed. The results of the study indicate the distribution and level of public services in Abha and Bisha. In Abha, there is a high level of public services, particularly in green areas, which contribute to improving the quality of services and meeting the entertainment needs of the population. In contrast, Bisha has a middling level of public services, likely due to its smaller population size and lesser focus on development as compared to Abha. The study also analyzes the differences in attitudes towards public services based on gender. The results indicate that there are no statistically significant differences between males and females in their perceptions of public services in both cities.

**Keywords:** smart cities; public services; data analysis; environmental sustainability

## 1. Introduction

Public service distribution is the process of providing essential services to the public, such as healthcare, education, transportation, and public safety. The distribution of these services is usually managed by government agencies or non-profit organizations responsible for making them accessible and useful to all members of society [1,2]. This may involve allocating resources, such as funding and human resources, according to the needs and priorities of different regions or communities. The efficient distribution of public services is critical to promoting social justice and improving the overall well-being of individuals and communities [2].

Smart city development has gained significant attention from researchers and policy-makers in recent years. Several studies have focused on different aspects of smart cities, including their impact on economic growth, environmental sustainability, and social well-being [3–5]. However, few studies have specifically examined the distribution of public services in smart cities.

One study by Alshehri et al. (2021) examined the challenges facing the implementation of smart city projects in Saudi Arabia. The study found that one of the main challenges is the lack of coordination between different government agencies responsible for providing public services. This lack of coordination can lead to inefficiencies in service delivery and duplication of efforts [4].

Another study by Alawadhi et al. (2020) focused on the role of technology in enhancing public service delivery in smart cities. The study found that technology can improve service delivery by providing real-time data on service demand and usage patterns. This data can be used to optimize service delivery and reduce wait times for citizens [3].

A third study by Bibri & Krogstie (2017) examined the concept of citizen-centric smart cities, where citizens are actively involved in shaping urban development policies and decision-making processes. The study found that citizen participation can lead to more efficient and effective service delivery, as citizens are better able to identify their needs and preferences [5].

This research uses a qualitative approach to assess public service distribution in smart cities in Saudi Arabia. The study involves a review of existing literature on smart city development and public service delivery. The study also includes interviews with government officials responsible for implementing smart city projects in Saudi Arabia.

In the realm of urban planning, the equitable distribution of public services within cities has emerged as a critical imperative, prompting extensive investigations in engineering, spatial studies, and statistical analyses. The effectiveness and fairness of service distribution play a pivotal role in urban planning aimed at enhancing the well-being and developmental prospects of city residents while simultaneously advancing environmental sustainability. To this end, the systematic and coordinated allocation of key public services constitutes a core tenet of urban planning, seeking to ensure widespread accessibility and equitable distribution throughout cities [6].

Abha and Bisha are two cities in southwestern Saudi Arabia. Abha, the capital of the Asir region, is known for its cool climate; Bisha, on the other hand, is a small town south of Asir known for its agricultural production, especially of dates and fruits. Both cities offer unique cultural experiences, including traditional markets, local cuisine and festivals celebrating their traditions.

The cities of Abha and Bisha in Saudi Arabia serve as the focal point of this comparative study, delving into the assessment of public service distribution within these urban centers. Situated in the Asir region in the southwest of the Kingdom, Abha and Bisha represent significant urban areas among the thirteen regions of Saudi Arabia. Abha, the capital of the Asir region, is located at the intersection of latitude $12'18°$ N and longitude $30'42°$ E, and is surrounded by Al-Hamr, Al-Sha'f, Marba, Khamis Mushait, Tayyib, and Al-Souda. Covering an approximate area of 290.7 km$^2$, Abha encompasses 44 residential neighborhoods, including notable ones such as Shamsan, Al-Manhal, Al-Andalus, Al-Azizia, Al-Salam, and Al-Khalidiyah, with a population of 446,697 people as of 2022 [7]. On the other hand, Bisha, located in the northwest of the Asir region, spans latitudes $0'19°$ to $51'20°$ N and longitudes $50'41°$ to $5'43°$ E. It shares borders with the Makkah Al-Mukarramah region to the north, Khaybar in Khamis Mushait Governorate to the south, Tathleeth, Jash, and Al-Sabikha in the Tathleeth Governorate to the east, and the Al-Baha region, Balqarn, and Al-Namas to the west. Bisha extends 185 km from north to south, with its width varying from east to west, ranging from 48 km in the south to 120 km in the north. It covers an area of 659 km$^2$, accommodating a population of 248,452 people as of 2022 [8]. The significance of this study area stems from Abha's status as the regional capital and a major tourism destination in the southern region of Saudi Arabia. Furthermore, the presence of green spaces and its central location within the Asir region have contributed to the investment of concentrated attention and development efforts in Abha. In contrast, Bisha, with its smaller population size, faces different challenges in terms of service provision and development, highlighting the need to examine public service distribution across these two cities in a comparative manner.

## 2. Research Goal

The present study aims to assess public service distribution in Abha and Bisha, shedding light on their similarities and differences in terms of service accessibility and satisfaction levels. By employing spatial modeling techniques and analyzing the satisfaction levels

of the population, this study seeks to investigate spatial justice and equality in service distribution, as well as the impact of public services on environmental sustainability. The study hypothesizes that there are no statistically significant differences between the two cities in terms of public services based on variables such as gender, age group, and category [9]. It also posits that public services moderately affect environmental sustainability and quality of life in Abha and Bisha, and that the population will express a high level of satisfaction with public services in both cities [10]. To achieve its objectives, the study employs a mixed-methods approach, combining the quantitative analysis of spatial data with qualitative assessments through surveys and interviews. The quantitative analysis involves the collection and examination of geospatial data describing the distribution of public services, including but not limited to healthcare facilities, educational institutions, transportation networks, and recreational spaces. The data was processed using Geographic Information System (GIS) techniques to generate maps and spatial indicators that provide a visual representation of the distribution patterns and spatial equity of public services in Abha and Bisha [11,12]. Additionally, to gain insights into the satisfaction levels and perceptions of the population regarding public services, a survey was conducted among a representative sample of residents in both cities. The survey questionnaire encompasses a range of factors, including accessibility, affordability, quality, and adequacy of public services. The collected data was analyzed using statistical methods, such as regression analysis and correlation tests, to determine the relationship between service provision and demographic characteristics, as well as to assess the impact of public services on environmental sustainability [13,14].

Complementing the quantitative analysis, qualitative methods such as interviews and focus group discussions were employed to gather in-depth perspectives from key stakeholders, including urban planners, policymakers, and community representatives. These qualitative insights will provide a nuanced understanding of the challenges, opportunities, and aspirations related to public service distribution in Abha and Bisha, helping to inform recommendations for improving service equity and sustainability in urban planning practices [15,16]. The findings of this study hold significant implications for urban planners, policymakers, and decision-makers involved in the development and management of cities. By identifying disparities in public service distribution, the study will contribute to evidence-based decision-making, enabling targeted interventions to address spatial inequalities and enhance the well-being of residents. Furthermore, the study's insights into the impact of public services on environmental sustainability can guide efforts to foster sustainable urban development and promote eco-friendly practices in Abha and Bisha [17,18]. While the study focuses on the specific contexts of Abha and Bisha, its methodology and findings can also serve as a reference for similar urban centers in Saudi Arabia and beyond. The comparative approach adopted in this study provides a framework for assessing public service distribution and spatial equity, which can be replicated and adapted to other cities, fostering a broader understanding of urban planning challenges and opportunities [19–21]. The study aims to assess the distribution of public services in Abha and Bisha, examining their impact on environmental sustainability and the satisfaction levels of the population. By employing a mixed-methods approach, the study seeks to generate valuable insights into spatial justice, service accessibility, and the relationship between public services and quality of life. The findings of this study have the potential to inform evidence-based decision-making in urban planning, contributing to more equitable and sustainable cities in Saudi Arabia and beyond [22–24].

Previous studies have shown that the distribution of public services is often unequal and can be influenced by factors such as income, race, and geography. For example, low-income communities may have less access to quality healthcare or education compared to wealthier neighborhoods. Similarly, rural areas may face challenges in accessing transportation or internet services compared to urban areas. Studies have also highlighted the importance of government policies and funding in addressing these disparities. Programs such as Medicaid and Head Start aim to provide healthcare and education [25–28].

### 3. Study Area

The study considers the cities of Abha and Bisha as applied fields. The two cities are located in the Asir region in the southwest of the Kingdom of Saudi Arabia, which is one of the thirteen regions in the Kingdom (Figure 1). The city of Abha is the capital of the Asir region and the headquarters of the emirate. It was named after the valley of the same name. The city of Abha is located at the confluence of latitude $12'18°$ N and longitude $30'42°$ E. A center in Al-Hamr (Abha Governorate), to the south are the centers of Al-Sha'f and Marba (Abha Governorate), to the east is the city of Khamis Mushait (Khamis Mushait Governorate), and to the west are the centers of Tayyib and Al-Souda (Abha Governorate).

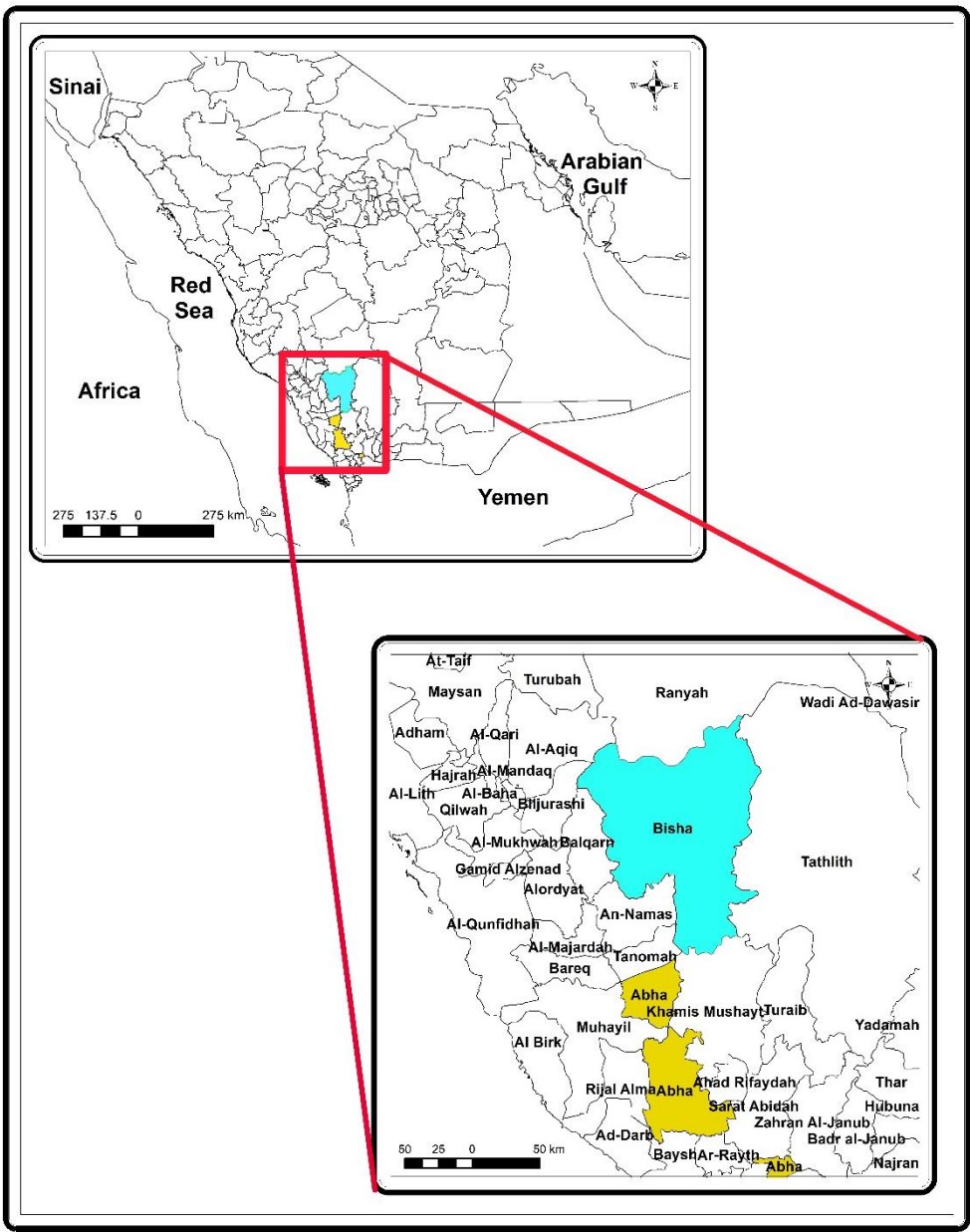

**Figure 1.** The location of the cities of Abha and Bisha in the Kingdom of Saudi Arabia as of 2023.

The total area of the city of Abha is about 290.7 km², distributed over 44 residential neighborhoods, the most important of which are Shamsan, Al-Manhal, Al-Andalus, Al-Azizia, Al-Salam, and Al-Khalidiyah, and the city housed a population of 446,697 people as of 2022 AD. As for Bisha, it is located in the northwest of the Asir region, between latitudes $0'19°$ and $51'20°$ N and longitudes $50'41°$ and $5'43°$ E. It is bordered on the north

by the Makkah Al-Mukarramah region, and to the south is the center of Khaybar in Khamis Mushait Governorate. It is bordered to the east by the centers of Tathleeth, Jash, and Al-Sabikha in the Tathleeth Governorate, and to the west by the Al-Baha region and the governorates of Balqarn and Al-Namas.

The city of Bisha extends from north to south with a length of 185 km, and its width varies from east to west; its lowest width was recorded in the south at about 48 km and the maximum width in the north at about 120 km. Its total area is 659 km$^2$, while its population was 248,452 people as of 2022 AD. Its importance is primarily due to its location on Wadi Bisha, the largest and most important valley in the Kingdom, which has the largest dam in the Kingdom, with a storage capacity of 325 million m$^3$. Bisha is famous for its agriculture, especially date palms, and the city of Bisha has a good network of roads connecting it to neighboring cities, the most important of which is the Khamis Mushait-Bisha-Raniyeh-Khurmah road, which is one of the essential roads linking the Asir region with the Makkah region.

The review of public services in smart cities has several theoretical implications related to key topics such as technology, governance, and citizen engagement. Firstly, the use of technology in public services has the potential to transform service delivery and improve efficiency. However, it is important to ensure that technology is used in a way that does not exacerbate existing inequalities or exclude certain groups of citizens.

Secondly, governance structures play a crucial role in ensuring that public services are delivered effectively and efficiently. Smart cities require new forms of governance that are more collaborative and inclusive, involving multiple stakeholders such as citizens, businesses, and government agencies.

Finally, citizen engagement is essential for the success of smart city initiatives. Citizens should be involved in the design and implementation of public services to ensure that they meet their needs and preferences. This requires a shift towards more participatory forms of decision-making that involve citizens in co-creating solutions.

From a practical perspective, the review highlights several consequences for managers and practitioners involved in delivering public services in smart cities. Firstly, they need to be aware of the potential benefits and risks associated with using technology in service delivery. They should also be prepared to work collaboratively with other stakeholders to develop effective governance structures.

## 4. The Hypotheses of the Study

To achieve the main objectives of the study, a number of hypotheses were formulated:

**Hypothesis 1.** *There are no statistically significant differences between the samples with regard to public services in the cities of Abha and Bisha when considering the variables of gender, age group, and class.*

**Hypothesis 2.** *Public services affect environmental sustainability and the quality of life in the cities of Abha and Bisha to a moderate degree.*

**Hypothesis 3.** *There is a high level of satisfaction within the population with regards to public services in the cities of Abha and Bisha.*

By testing these hypotheses, it was possible to arrange the study elements along two main axes:

First axis: statistical analysis of the study sample trends related to public services.

Second axis: spatial modeling of the distribution of level of satisfaction within the population with regards to public services.

### 5. Methodology

The nature of the study requires the use of an analytical descriptive approach in addition to three other approaches, namely, a substantive approach, regional approach, and historical approach. It also used different statistical methods via the SPSS V-20 statistical software package:

- Five-point Likert scale
- Arithmetic mean in terms of the reference between scores
- Standard deviations to infer the dispersion and variance of grades
- Kolmogorov–Smirnov one-sample test to check member distribution efficiency
- Mann–Whitney U test to measure differences between two independent samples
- Kruskal–Wallis test to measure the differences between several independent samples

The researcher depended on Geographic Information System (GIS) to produce maps and analyzed them spatially using a program called "Arc GIS 10.8" in order to explain the spatial modeling. This step was achieved by creating a geo-database using a program called "Arc Catalog". The geo-database comprises a set of feature classes that include information on different features of the city, and thereby enables the researcher to conduct spatial statistical analyses.

The data collected from the literature review and interviews were analyzed. The analysis identifies common themes related to public service distribution in smart cities in Saudi Arabia. The themes were then used to identify gaps in current practices and suggest recommendations for improving public service delivery.

General questionnaires were used to collect data from participants. They consisted of a series of questions designed to gather information about the subjects in question. The questionnaire of this study was designed to be administered in a variety of ways, including online, in person, or by mail and was designed to ensure that the questions were clear, unbiased, and that the information needed for the research could be efficiently obtained. The questions are open/closed and also include multiple choice for scale scoring. A small group of participants was pre-tested before using the questionnaire in the main study.

The study population comprises citizens and residents of different sexes and ages in the cities of Abha and Bisha, where the questionnaire was applied to a sample of the research community, which garnered 479 responses in the city of Abha and 496 responses in the city of Bisha, all of which are correct, complete, and valid for analysis, as shown in Table 1.

**Table 1.** The number and percentage of the study sample groups in the cities of Abha and Bisha.

| Variables | Variable Category | Bisha | | Abha | |
|---|---|---|---|---|---|
| | | **N** | **%** | **N** | **%** |
| Gender | Male | 274 | 55.2% | 276 | 57.6% |
| | Female | 222 | 44.8% | 203 | 42.4% |
| | **Total** | **496** | **100%** | **479** | **100%** |
| Age | Less than 20 years | 130 | 26.2% | 203 | 42.4% |
| | Between 20 and 40 years | 231 | 46.6% | 177 | 37.0% |
| | More than 40 years | 135 | 27.2% | 99 | 20.7% |
| | **Total** | **496** | **100%** | **479** | **100%** |
| Category | Citizen | 329 | 66.3& | 377 | 78.7% |
| | Resident | 167 | 33.7% | 102 | 21.3% |
| | **Total** | **496** | **100%** | **479** | **100%** |

Description of the study sample:

As shown in Figure 2, the characteristics of the study sample individuals were identified according to the following variables:

1.    Gender: (Male, Female).
2.    Age: (less than 20 years, between 20 and 40 years, and more than 40 years).
3.    Category: (Citizen, Resident).

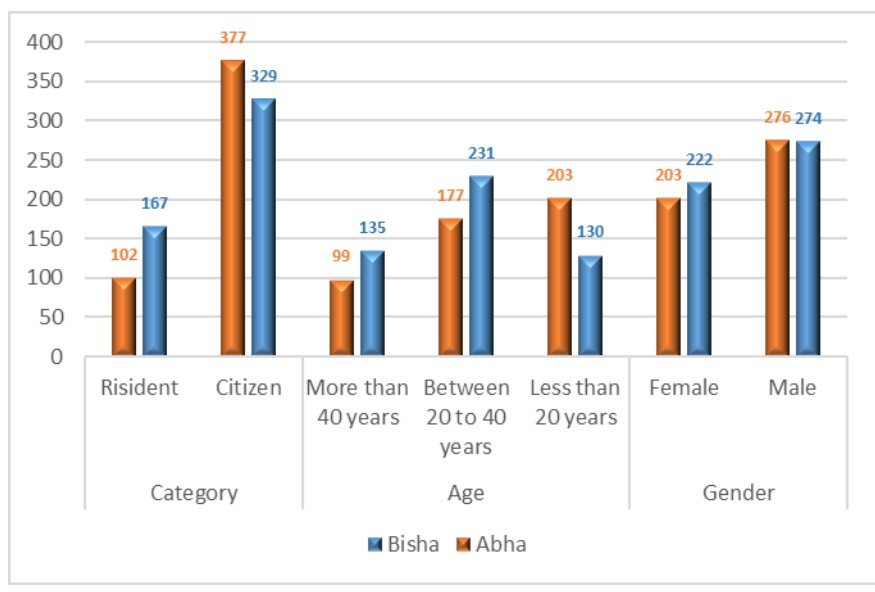

**Figure 2.** The numerical and relative distribution of the study sample groups in the cities of Abha and Bisha.

Normality of the distribution of the study sample:

To check the distribution among the categories of the research sample, the Kolmogorov–Smirnov one-sample test was used to see if the data obtained from the sample followed a normal distribution or not, in order to determine which tests would be appropriate for each case: teachers' tests or non-parametric tests—parametric tests were used when the distribution was normal with a significance level greater than 0.05, while non-parametric tests were used when the distribution was abnormal with a significance level less than 0.05. The test results are as shown in the following Table 2:

**Table 2.** Results of the Kolmogorov–Smirnov one-sample test to examine the evenness of distribution in the research sample.

|  | Abha | | Bisha | |
|---|---|---|---|---|
|  | Kolmogorov–Smirnov Z | Sig. | Kolmogorov–Smirnov Z | Sig. |
| Gender | 8.325 | 0.000 | 8.199 | 0.000 |
| Age | 5.928 | 0.000 | 5.196 | 0.000 |
| Category | 10.623 | 0.000 | 9.465 | 0.000 |

It is clear from the results of the Kolmogorov–Smirnov one-sample test, which examines the moderation of the distribution of the sample members, that there is no moderation between the distribution of the sample members in the cities of Abha and Bisha in all research variables, and, accordingly, the Mann–Whitney U test was used for two samples. Two independent samples with the variables of gender and category were used, and the Kruskal–Wallis test was used for several independent samples with the variable of age, as shown later.

## 6. Results and Discussion

The outcome of public service allocation depends on various factors such as government policy, budget allocation, and implementation strategy. When public services are distributed efficiently and equitably, it can improve citizens' quality of life, boost economic growth, and reduce inequality. On the other hand, when public services are not distributed fairly and efficiently, it leads to social unrest and citizen dissatisfaction. Therefore, the government must prioritize the distribution of public services to ensure that public services benefit all sectors of society.

The statistical analysis of the trends within the study sample towards public services follows.

The researcher calculated the frequencies, percentages, arithmetic averages, and standard deviation of the sentiments surrounding green spaces in the cities of Abha and Bisha and arranged the arithmetic averages in descending order to determine the most commonly expressed sentiments. The results are shown in Table 3.

**Table 3.** Arithmetic means and standard deviations of the axes and sentiments surrounding green spaces and their impact on air pollution in the cities of Abha and Bisha.

| | | | Strongly Disagree | Disagree | Neutral | Agree | Strongly Agree | Mean | Std. Deviation | Arrangement | Class |
|---|---|---|---|---|---|---|---|---|---|---|---|
| | | | | | Scale | | | | | | |
| | **ABHA** | | | | | | | | | | |
| 1 | Green spaces have been expanded in order to improve the quality of services provided in Saudi cities. | N | **40** | **105** | 60 | 114 | 160 | 3.52 | 1.364 | 3 | Agree |
| | | % | 8.4 | 21.9 | 12.5 | 23.8 | 33.4 | | | | |
| 2 | Green spaces have been expanded to improve the urban landscape in Saudi cities. | N | 36 | 99 | 77 | 109 | 158 | 3.53 | 1.333 | 2 | Agree |
| | | % | 7.5 | 20.7 | 16.1 | 22.8 | 33 | | | | |
| 3 | One of the advantages of the expansion of green spaces is the development and diversification of entertainment opportunities to meet the needs of the population. | N | 33 | 83 | 61 | 118 | 184 | 3.70 | 1.319 | 1 | Agree |
| | | % | 6.9 | 17.3 | 12.7 | 24.6 | 38.4 | | | | |
| | **Total: Improving Public Utilities and Services** | | | | | | | 3.59 | 0.850 | | Agree |
| | **BISHA** | | | | | | | | | | |
| 1 | Green spaces have been expanded in order to improve the quality of services provided in Saudi cities. | N | 98 | 108 | 92 | 96 | 102 | 2.99 | 1.424 | 3 | Neutral |
| | | % | 19.8 | 21.8 | 18.5 | 19.4 | 20.6 | | | | |
| 2 | Green spaces have been expanded to improve the urban landscape in Saudi cities. | N | 92 | 103 | 116 | 88 | 97 | 2.99 | 1.383 | 2 | Neutral |
| | | % | 18.5 | 20.8 | 23.4 | 17.7 | 19.6 | | | | |
| 3 | One of the advantages of the expansion of green spaces is the development and diversification of entertainment opportunities to meet the needs of the population. | N | 98 | 74 | 91 | 109 | 124 | 3.18 | 1.460 | 1 | Neutral |
| | | % | 19.8 | 14.9 | 18.3 | 22 | 25 | | | | |
| | **Total: Improving Public Utilities and Services** | | | | | | | 3.05 | 0.834 | | Neutral |

The multiplicity of public services provided in the city of Abha, especially in terms of green spaces, as a total measure of public services in the city of Abha obtained a score of Agree, with an arithmetic mean of 3.59. The total for all the phrases obtained a score of Agree, with a small standard deviation of 0.850, which indicates agreement among the opinions of the research sample with the phrases as a whole. This indicates the multiplicity of public services in the city of Abha, especially in terms of green spaces, which can be explained as resulting from the fact that it constitutes the headquarters of the emirate, wherein about half of the population of the Asir region is concentrated, in addition to the fact that it represents the first tourist destination in the south of the Kingdom of Saudi Arabia.

The statistical results were different in the city of Bisha, where the total public services scale obtained a Neutral score, with an arithmetic mean of 3.05. The total for all the phrases obtained a Neutral score, with a small standard deviation of 0.834, which indicates agreement among the opinions of the sample. The results for the phrases as a whole indicate a lack of public services in the city of Bisha, due to the city's small population compared to the city of Abha, in addition to the latter receiving the greatest amount of attention in the Asir region due to its tourist nature.

Phrase No. 3, which reads "One of the advantages of the expansion of green spaces is the development and diversification of entertainment opportunities to meet the needs of the population." had the highest average score in the two cities of Abha and Bisha, with a score of 3.70 in the first, and 3.18 in the second, which indicates the importance of green spaces and their vital role in achieving environmental sustainability.

Population attitudes towards public services by gender follow.

The study used the Mann–Whitney U test for two independent samples (Male, Female) to measure the differences between the opinions of the members of the study sample towards public services in the cities of Abha and Bisha according to the gender variable. By analyzing the numbers in Table 4, it was found that there were no significant differences at the significance level of $\propto = 0.05$ between the averages of the responses of the research sample individuals in the cities of Abha and Bisha in terms of public services according to the variable of gender, where the level of significance reached an average value of 0.728 in Abha, while an average of 0.808 was recorded in the city of Bisha, which are both values greater than 0.05. This indicates that the distribution of services in the two cities was not perceived differently by males and females.

**Table 4.** The results of the Mann–Whitney U test for two independent samples to detect differences between the averages of the study sample responses in the cities of Abha and Bisha according to difference in gender.

| | Gender | N | Mean Rank | Sum of Ranks | Mann–Whitney U | Sig. |
|---|---|---|---|---|---|---|
| | | | **ABHA** | | | |
| | **Male** | **276** | **241.88** | **66,757.50** | 27,496.500 | 0.728 |
| | Female | 203 | 237.45 | 48,202.50 | | |
| Total: Improving Public Utilities and Services | | | **BISHA** | | | |
| | Male | 274 | 249.90 | 68,472.00 | 30,031.000 | 0.808 |
| | Female | 222 | 246.77 | 54,784.00 | | |

Population attitudes towards public services according to age group follow (Table 5).

**Table 5.** The results of the Kruskal–Wallis test for several independent samples reveal the differences between the averages of the study sample responses in the cities of Abha and Bisha according to the age variable.

| | Age | N | Mean Rank | Chi-Square | Sig. |
|---|---|---|---|---|---|
| | **ABHA** | | | | |
| | Less than 20 years | 203 | 253.60 | | |
| Total: Improving Public Utilities and Services | Between 20 and 40 years | 177 | 255.36 | 20.279 | 0.000 |
| | More than 40 years | 99 | 184.66 | | |
| | **BISHA** | | | | |
| | Less than 20 years | 130 | 265.22 | | |
| | Between 20 and 40 years | 231 | 255.90 | 7.946 | 0.019 |
| | More than 40 years | 135 | 219.73 | | |

The non-parametric Kruskal–Wallis test was used for several independent samples in the categories of less than 20 years, between 20 and 40 years, and more than 40 years, to measure the differences between the study samples in environmental sustainability in the two cities of Abha and Bisha according to the age variable.

- The opinions of the study sample towards public services in the city of Abha was affected according to the variable of age. Statistically significant differences were observed at the level of significance of $\propto = 0.05$, as its value reached 0.000, which is a value smaller than 0.05, indicating the presence of statistically significant differences between the average responses of the research sample and the category of between 20 and 40 years. This can be explained by the fact that that age group is more mobile in nature, and is therefore the category most affected by the distribution of public services in the city of Abha.
- The opinions of the study sample towards public services in the city of Bisha was affected according to the variable of age. Statistically significant differences were observed at the level of significance of $\propto = 0.05$, as its value reached 0.019, which is a value smaller than 0.05, indicating the presence of statistically significant differences between the average responses of the research sample individual and the category of less than 20 years. This indicates that this category does not enjoy the required services, especially with regard to green spaces, which constitute a haven for that category to practice various recreational activities.

Population attitudes towards public services according to nationality follow.

The study used the Mann–Whitney U test for two independent samples (citizen and resident) to measure the differences between the opinions of the members of the study sample regarding green spaces in the cities of Abha and Bisha according to the category variable. By analyzing the numbers in Table 6, the following facts can be observed:

- There are statistically significant differences at the significance level of $\propto = 0.05$ between the averages of the sample responses in the cities of Abha and Bisha. It reached 0.031 in the city of Abha, while it reached 0.000 in the city of Bisha, which are both smaller values than 0.05, indicating that there are statistically significant differences between the average responses of the research sample members in the two cities attributed to the category variable in favor of the Citizen category. This indicates that citizens are more affected by the distribution of public services, as citizens are keen to benefit from those services, in contrast to residents who are more interested in work and getting the most rest without concern for enjoying public services, which is especially impacted by the prevalence of singles among residents.

**Table 6.** The results of the Mann–Whitney U test for two independent samples reveal the differences between the averages of the study sample responses in the cities of Abha and Bisha according to the category variable.

|  | Category | N | Mean Rank | Sum of Ranks | Mann–Whitney U | Sig. |
|---|---|---|---|---|---|---|
|  | **ABHA** | | | | | |
| Total: Improving Public Utilities and Services | Citizen | 377 | 232.94 | 87,818.50 | 16,565.500 | 0.031 |
|  | Resident | 102 | 266.09 | 27,141.50 | | |
|  | **BISHA** | | | | | |
|  | Citizen | 329 | 230.28 | 75,763.00 | 21,478.000 | 0.000 |
|  | Resident | 167 | 284.39 | 47,493.00 | | |

The results of measuring the differences between the average response of the population sample in the two cities follow.

The researcher relied on the Mann–Whitney U test for two independent samples (Abha and Bisha) to measure the differences between the opinions of the individuals in the two samples towards public services. By analyzing the numbers in Table 7, the following results can be observed:

– There are statistically significant differences between the averages of the responses of the study sample in the city of Abha and the research sample in the city of Bisha. The level of significance was 0.003, which is a smaller value than 0.05, which indicates the existence of significant differences. Statistical significance at the level of significance of $\propto = 0.05$ between the average responses of the research sample in the city of Abha and the research sample in the city of Bisha according to the variable of City, demonstrated a higher score in the city Abha. This is due to the concentrated population in the city of Abha, which impacts the availability of services. It is the seat of the Emirate, the capital of the Asir region, and the first domestic tourist destination in the south of the Kingdom of Saudi Arabia.

**Table 7.** The results of the Mann–Whitney U test for two independent samples to detect differences between the averages of the study sample responses in the cities of Abha and Bisha.

|  | City | N | Mean Rank | Sum of Ranks | Mann–Whitney U | Sig. |
|---|---|---|---|---|---|---|
| Total: Improving Public Utilities and Services | Abha | 479 | 460.85 | 220,748.50 | 2.982 | 0.003 |
|  | Bisha | 496 | 514.22 | 255,051.50 | | |

## 7. Spatial Modeling of Public Service Distribution

Monitoring the levels of residents' satisfaction with public services contributes to identifying the level of services provided in the two cities, determining what needs to be developed and what must be maintained at its current level in a way that ensures the continued satisfaction of residents and increases the two cities' populations and attractiveness to tourists and thus achieves environmental sustainability.

The researcher relied on Arc GIS 10.8, which is the most important GIS 10.8 program, to carry out the process of spatial modeling of public services in the cities of Abha and Bisha. The model displays the public services in the two cities by showing the areas that completely or partially intersect with the public services using the Intersect Overlay process, which is used to obtain a set of common data between two overlapping layers by converting it into connected Raster data through the Spatial Analyst menu, from which the Interpolate to Raster command was selected, including Inverse Distance Weighted (IDW). Then, Cell Statistic was used for the layers of standards, taking the average values for the mean, then reclassifying the data to divide it into main categories according to the concentration public services. By analyzing Figure 3, the following facts can be observed:

– There is a low percentage of satisfaction among the population sample with public services in the cities of Abha and Bisha, as it reached about 6% of the total distribution of levels of satisfaction in the two cities. Despite this, the city of Abha is superior to the city of Bisha in terms of population satisfaction by more than four-fifths of that percentage, which can be explained by the multiplicity of services and green spaces in the city of Abha compared to the city of Bisha.

– There is a direct positive correlation, with a value of +0.91, between the distribution of vegetation cover and green spaces on the one hand, and the levels of population satisfaction on the other hand.

– There is an expansion of the areas that witnessed a "neutral" level of satisfaction from the population, as they constitute 21.2% of the total area of the two cities, and are concentrated in the city of Abha compared to the city of Bisha, especially in the areas adjacent to the city center.

– There is a wide amount of areas that witnessed dissatisfaction with public services in the cities of Abha and Bisha, as they cover an area estimated at about 66% of the total area of the two cities, and the levels of dissatisfaction are clearly concentrated in the city of Bisha, especially in the northern and eastern parts of the city. This may be explained by the lack of public services areas, especially green spaces, and the vegetation cover in those part. The areas of dissatisfaction are concentrated in the northeastern parts of the city of Abha.

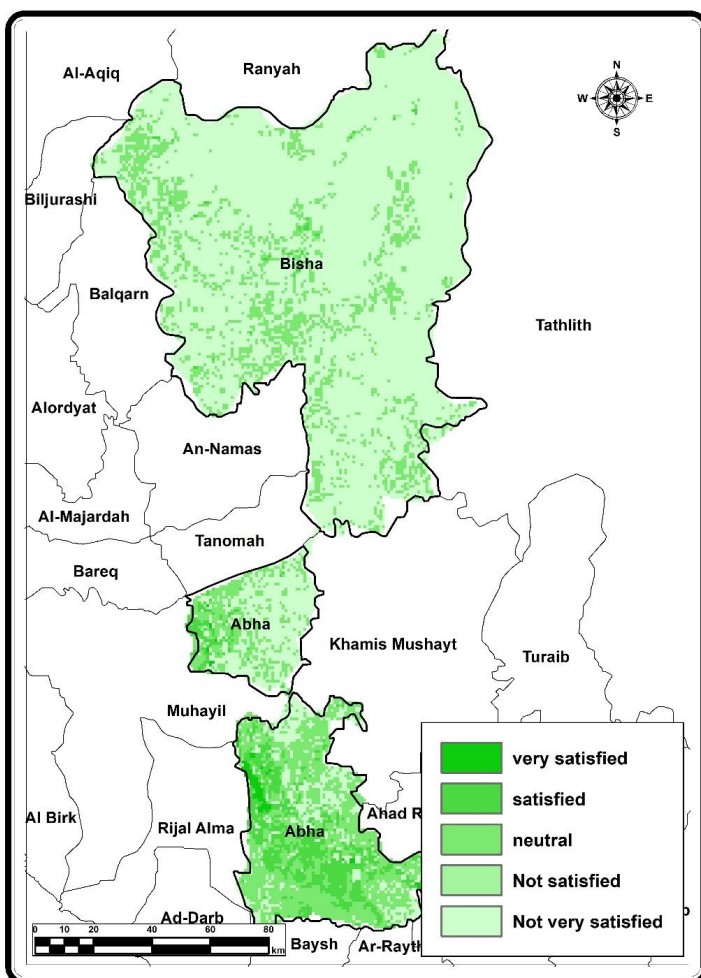

**Figure 3.** Spatial modeling of the distribution of public services related to green spaces and its relationship to the distribution of vegetation cover in the cities of Abha and Bisha in 2023. Source: prepared by the researcher based on Arc GIS 10.8.

## 8. Conclusions

The distribution of public services depends on various factors such as government policies, budget allocation, and implementation strategies. If public services are distributed efficiently and equitably, it can lead to improved quality of life for citizens, increased economic growth, and reduced inequality. On the other hand, if public services are not distributed fairly or effectively, it can lead to social unrest and dissatisfaction among citizens. Therefore, it is essential for governments to prioritize the distribution of public services and ensure that they reach all segments of society.

This study puts forward several conclusions and recommendations, which can be summarized as follows: the multiplicity of public services provided in the city of Abha, especially green spaces, on the total scale of satisfaction with public services in the city of Abha, obtained a score of Agree, with an arithmetic mean of 3.59, while the city of Bisha obtained a score of Neutral, with an arithmetic mean of 3.05. The total score of agreement with all the phrases was Neutral, which indicates a lack of public services in the city of Bisha. There are no statistically significant differences between the averages of the satisfaction responses in terms of public services in the cities of Abha and Bisha of the research sample individuals according to the gender variable, as the level of significance was recorded at an average value of 0.728 in Abha, while an average of 0.808 was recorded in the city of Bisha. There was an effect on the satisfaction with public services in the cities of Abha and Bisha of the study sample according to the age variable, as statistically significant differences were detected at the level of significance of $\propto = 0.05$. Its value amounted to 0.000, demonstrating favor in the category of between 20 and 40 years in the city of Abha, while its value was 0.019 in the city of Bisha, demonstrating favor in the category of less than 20 years. There are statistically significant differences attributed to the Category variable between the averages of the respondents' responses in the cities of Abha and Bisha, as the significance value amounted to 0.031 in the city of Abha and reached 0.000 in the city of Bisha, demonstrating favor in the Citizen category. There are statistically significant differences between the study samples according to the City variable in the cities of Abha and Bisha in terms of satisfaction towards public services, as it amounted to 0.003, demonstrating favor in the city Abha, which is due to the population concentration in the city of Abha and the availability of various public services in it, as it is the headquarters of the Emirate and the capital of the Asir region. There is a low percentage of the population sample which is satisfied with public services in the cities of Abha and Bisha, as it reached about 6% of the total distribution of levels of satisfaction in the two cities, with a positive direct correlation with a value of +0.91 between the distribution of vegetation cover and the levels of satisfaction of the population. There is a wide amount of areas that witnessed dissatisfaction with public services in the cities of Abha and Bisha, as they cover an area estimated at about 66% of the total area of the two cities. The levels of dissatisfaction are clearly concentrated in the city of Bisha, especially in the northern and eastern parts of the city, which may be explained by the lack of public services areas, especially green spaces, and the vegetation cover in those parts, while in Abha the areas of dissatisfaction are concentrated in the northeastern parts of the city.

According to the results of this study, it was found that the cities of Abha and Bisha need more improvement to and distribution of public services. This can be achieved by considering factors such as further strengthening coordination among service providers, increasing investment in, and improving public services along with quality of service and provision of more facilities and ease of access to these facilities to meet the expected growth in the number of residents in the coming years.

In terms of future areas of study, different ideas can be explored to find further evidence for the development of smart cities, such as the following:

Firstly, research on the impact of smart city technology on public service distribution can explore the impact of smart city technology on public service distribution in Saudi Arabia. It can analyze the effectiveness of smart city technology in improving the distribution of public services such as healthcare, education, transportation, and utilities.

Secondly, research on citizen satisfaction with public service distribution can investigate citizen satisfaction with public service distribution in smart cities in Saudi Arabia. It can examine the factors that influence citizen satisfaction, such as accessibility, quality, and affordability of public services.

Finally, research on equity and accessibility of public services can focus on the equity and accessibility of public services in smart cities in Saudi Arabia. It can analyze how smart city technology can improve access to public services for marginalized communities and reduce disparities in service delivery.

**Funding:** The Deanship of Scientific Research at the University of Bisha funded this research.

**Informed Consent Statement:** Not applicable.

**Data Availability Statement:** Not applicable.

**Acknowledgments:** The author is thankful to the Deanship of Scientific Research at University of Bisha for supporting this work through the Fast-Track Research Support Program.

**Conflicts of Interest:** The author declares no conflict of interest.

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
