# Peer review of "Assessing Public Service Distribution in Abha and Bisha Cities, Saudi Arabia: A Comparative Study"

_sustainability, doi:10.3390/su151310343_

Round 1

Reviewer 1 Report

This work is interesting but some of the major points should be considered which are given below:

-          The Introduction is not described enough in detail.

-          The literature is very poor for scientific article. I miss relevant literature from the field.

-          The methodology is not demanding enough.

-          Table 3 should be better explained.

-          The results comparing all different tests using are missing.

-          I recommend rewriting the conclusion.  The concluding remarsk should be more specific and better explained. The further study is mentioned, but it should be more concretized.

In summary, the article is sufficiently interesting, but I recommend rejecting it, otherwise, please, follow all the comments above.

Author Response

Reviewer-1

  1. The Introduction is not described enough in detail.

Response: Thank. The introduction of scientific article has been re described and part of missing detail being added as required

  1. The Literature is very poor for scientific article. Missing relevant literature from the field.

Response: Thanks, A number of different literatures related to the research objectives have been added. Through the narrative, the necessary comparisons are made.

  1. The Methodology is not demanding enough.

Response: Thank. The methodology of article paper has been rewriting as required

  1. Table-3 should be better explained.

Response: Thank. More explanation added to part regarding the Table 3 through the literature discussed.

  1. Rewriting the conclusion.

Response: Thank. The conclusion of article paper has been rewriting and auditing as required

Reviewer 2 Report

NA

Author Response

Reviewer-2

  1. Provide more details about the questionnaire used for data collection, including the sample size, sampling method, and any measures taken to ensure the questionnaire's reliability and validity.

Response: Thanks, All steps that have been made are discussed regarding the questionnaire including the data collection, sampling sizes and analysis the results that have been achieved.

  1. The results regarding the high level of public services in Abha and the neutral level in Bisha are insightful. However, it would be helpful to provide specific examples or data points to support these findings.

Response: Thanks, Specific examples regarding the high level of public services have been added to support the finding.

  1. While the paper mentions the absence of statistically significant differences in attitudes towards public services between genders, it would be beneficial to explore the potential reasons behind this finding or discuss any existing literature on the topic.

Response: Thanks, Enhancement validity have been done by comparing multiple cases, enhancement for the validity of findings. This is because the difficult control for confounding variables that may be present in a single case study.

  1. Include recommendations or policy implications based on the study's findings. How can the insights from this research contribute to improving public service distribution in Abha, Bisha, or similar cities?

Response: Thanks, Recommendation part have been added to the section of results as required.

  1. Review the paper for clarity, grammar, and formatting to ensure it meets scholarly standards.

Response: Thank. The language of scientific article has been edited and some part being rewritten as required

Reviewer 3 Report

1. What are your contributions to the paper? (highlight them)

2. What are the merits compared to the existing approach? (prepare a comparison table)

3. English language and style are fine/minor spell check is required.

5. Auther needs some more relevant literature reviews. prepare a summary table of it to improve the quality of the article.

6. lack of relevant references (add more)

Minor editing of the English language is required.

Author Response

Reviewer-3

  1. What are your contributions to the paper?

 Response: Thanks. This article provides valuable insights into the distribution of public services in two Saudi Arabian cities. The study highlights the importance of equitable distribution of public services to ensure social and economic development in urban areas. Policymakers can use the findings of this study to improve the allocation of resources and improve the quality of public services in these cities. Furthermore, this paper contributes to the literature on public service delivery in developing countries and provides a framework for future research on the topic.

  1. What are the merits compared to the existing approach.

 Response: Thanks.

  1. Providing a comprehensive understanding through a comparative study allows researchers to compare and contrast different aspects of a phenomenon across different contexts. This provides a more comprehensive understanding of the phenomenon under investigation.
  2. Enables identification of similarities and differences by comparing different cases, identification similarities and differences in the distribution of public services. That can help policymakers to identify best practices and areas for improvement.
  3. English language and style are fine/minor spell check is required.

 Response: Thank. The author always strive to produce grammatically correct and coherent sentences. What was requested has been done. I am fully prepared if you find any other notes

  1. Author needs some more relevant literature review, Prepare summary table of it to improve the quality of article    

Response: Thanks, A number of different literatures related to the research objectives have been added. Through the narrative, the necessary comparisons are made.

  1. Lack of relevant references

Response: Thanks, The number of references related to the research has been added.

6- Minor editing of the English language is required 

Response: Thank. The language of scientific article has been edited and some part being rewritten as required

Reviewer 4 Report

Dear Author(s),

I am pleased to have the opportunity to review your manuscript titled "Comparative Study for Assessing Public Service Distribution in Abha and Bisha Cities." I have some suggestions that I believe will help improve the quality of this version:

The introduction is a crucial section of an academic article and could benefit from some improvements. I recommend placing greater emphasis on the research gap you are addressing. Please identify and refer to other articles in the same research domain to define this gap clearly.

Furthermore, I suggest emphasizing the originality of your work in this section. I recommend using relevant keywords to search major academic databases such as Scopus, Web of Science, and Google Scholar. This will allow you to distinguish your article significantly from existing literature.

In the hypothesis development section, defining the research streams related to the topic would be desirable and highlighting how your manuscript contributes to them. As a suggestion, you may refer to the following paper:

Crane, A., Matten, D., & Spence, L. J. (2013). Corporate social responsibility in a global context. Chapter in: Crane, A., Matten, D., and Spence, LJ, 'Corporate Social Responsibility: Readings and Cases in a Global Context, 2, 3-26.

The methodology section requires strengthening. I recommend referencing previous literature that has employed the same methodology. Additionally, it should be clear why the chosen method is the most suitable for obtaining the desired results. In this regard, please provide more details on the characteristics of your research design.

The presentation of the results is clear, and I appreciate that.

The implications section needs more attention, particularly regarding theoretical implications related to the key topics discussed in the review. Additionally, the practical consequences should serve as a meaningful stimulus for managers and practitioners. Unfortunately, this aspect is currently lacking in your work.

It would be beneficial to include a section on future lines of research. I suggest using the following text as a basis for developing future research ideas that can be useful for other scholars:

• https://doi.org/10.1016/j.jbusres.2022.06.011

Thank you for your hard work, and I wish you the best in your future research endeavours.

I suggest some editing regarding language clarity.

Author Response

Reviewer-4

  1. The introduction is a crucial section of an academic article and could benefit from some improvements. I recommend placing greater emphasis on the research gap you are addressing. Please identify and refer to other articles in the same research domain to define this gap clearly. 

Response: Thank. The introduction of scientific article has been re described and part of gap research being added as required.

  1. Furthermore, I suggest emphasizing the originality of your work in this section. I recommend using relevant keywords to search major academic databases such as Scopus, Web of Science, and Google Scholar. This will allow you to distinguish your article significantly from existing literature.

Response: Thank. The keyword have been updated as required.

  1. The methodology section requires strengthening. I recommend referencing previous literature that has employed the same methodology. Additionally, it should be clear why the chosen method is the most suitable for obtaining the desired results. In this regard, please provide more details on the characteristics of your research design

Response: Thank. The methodology of article paper has been reviewed as required with more details on the characteristics of the research design.

  1. The implications section needs more attention, particularly regarding theoretical implications related to the key topics discussed in the review. Additionally, the practical consequences should serve as a meaningful stimulus for managers and practitioners. Unfortunately, this aspect is currently lacking in your work.

Response: Thank. The implications section have been added to explain the regarding theoretical implications related to the key topics discussed in the review. 

  1. It would be beneficial to include a section on future lines of research. I suggest using the following text as a basis for developing future research ideas that can be useful for other scholars.

Response: Thank. The future line section have been added to support the research result that have been gained.

6- I suggest some editing regarding language clarity.

Response: Thank. The language of scientific article has been edited and some part being rewritten as required.

Round 2

Reviewer 1 Report

I agree with the revised version. 

Minor editing of English language required.

Reviewer 4 Report

Dear authors,

The paper appears improved. However, I suggest proofreading by a native speaker. In this sense, the text has some opportunities for improvement.

I wish you all the best in your future research.

I consider the paper greatly improved. However, the paper requires language revision, probably professional proofreading.